# Structure sensitivity in gas sorption and conversion on metal-organic frameworks

Guusje Delen [1,2], Matteo Monai [1,2], Katarina Stančiaková[1,2], Bettina Baumgartner [1], Florian Meirer [1] & Bert M. Weckhuysen [1] ✉

Many catalytic processes depend on the sorption and conversion of gaseous molecules on the surface of (porous) functional materials. These events often preferentially occur on specific, undercoordinated, external surface sites. Here we show the combination of in situ Photo-induced Force Microscopy (PiFM) with Density Functional Theory (DFT) calculations to study the site-specific sorption and conversion of formaldehyde on the external surfaces of well-defined faceted ZIF-8 microcrystals with nanoscale resolution. We observed preferential adsorption of formaldehyde on high index planes. Moreover, in situ PiFM allowed us to visualize unsaturated nanodomains within extended external crystal planes, showing enhanced sorption behavior on the nanoscale. Additionally, on defective ZIF-8 crystals, structure sensitive conversion of formaldehyde through a methoxy- and a formate mechanism mediated by Lewis acidity was found. Strikingly, sorption and conversion were influenced more by the external surface termination than by the concentration of defects. DFT calculations showed that this is due to the presence of specific atomic arrangements on high-index crystal surfaces. With this research, we showcase the high potential of in situ PiFM for structure sensitivity studies on porous functional materials.

The performance of a functional material is often determined by only a small percentage of its surface sites and their atomic configuration. This phenomenon, known as structure sensitivity, describes the relationship between the fraction of exposed crystal surfaces and the rate of conversion and is well known in the field of heterogeneous catalysis[1–3]. For example, in supported metal nanoparticle catalysts, the variation of metal nanoparticle size results in the exposure of a different fraction of active surface sites, leading to size-dependent performance[4].

Analogously, it is often observed for porous functional materials, such as metal-organic frameworks (MOFs), that their functionality is strongly dependent on their outer surface, despite their inner porosity[5]. As a result, evidence for the structure sensitivity of porous functional materials can be found in the literature[6,7]. For example, Pang and coworkers showed by ex situ scanning microscopy (SEM) analysis that when ZIF-8 crystals were exposed to acidic $SO_2$ gas, the external surface of the {100} facet was more stable than the {110} facet[8]. However, such an ex situ technique is not able to describe the guest–host interaction during gas exposure. Additionally, the performance of undercoordinated crystal surface edges and corners was not considered[8], despite the fact that these high-energy crystal planes often present highly important sorption, and/or conversion, sites on functional materials[9,10].

Undercoordinated sites are purposely introduced in defect engineering approaches, for example, through the incorporation of defective linkers[11]. The resulting undercoordinated metal centers can behave as Lewis acid sites with potentially increased activity towards gas sorption and/or conversion[12]. The localized integration of defects can be characterized by highly sensitive techniques, such as high-resolution transmission electron microscopy (HRTEM), scanning electron diffraction (SED), or atom probe tomography (APT)[13–17]. However, to study the nanoscale guest–host interaction between

[1]Inorganic Chemistry and Catalysis, Institute for Sustainable and Circular Chemistry and Debye Institute for Nanomaterials Science, Utrecht University, Utrecht, The Netherlands. [2]These authors contributed equally: Guusje Delen, Matteo Monai, Katarina Stančiaková. ✉e-mail: b.m.weckhuysen@uu.nl

(defective) external surface sites of functional porous materials, surface-sensitive in situ techniques are required[18–21].

Infrared spectroscopy is an ideal technique to describe guest–host interactions[22], but it suffers from low spatial resolution. Moreover, the low ratio of crystallite surface vs. bulk atoms may result in a loss of information on the external surface performance of porous functional materials. Tip-enhanced Raman spectroscopy (TERS) and photo-induced force microscopy (PiFM), are AFM-based vibrational techniques able to circumvent these limitations, with a spatial resolution down to the nanoscale (e.g., 5 nm for PiFM)[23,24]. Both techniques are non-destructive, label-free, can be performed in situ, and can be used to study a broad range of materials[25]. However, TERS suffers from arduous tip design- and preparation protocols, as well as measurement stability issues, which are not encountered when using PiFM. We recently showcased how PiFM can be used to study guest–host interactions in situ, looking at water sorption and induced defect formation on the surface of archetypical MOF thin films[26]. However, studying structure sensitivity requires the use of well-defined systems, where different crystal surface terminations can be imaged simultaneously under the same conditions.

In this work, we used in situ PiFM, in combination with density functional theory (DFT) calculations, to unravel structure sensitivity in gas sorption and conversion on the external surface of a microcrystalline MOF material in unprecedented detail (Fig. 1). Here, we show that with this toolbox we can purposely study highly functional, external surfaces of MOF crystal planes during gas exposure, which is not possible through conventional, bulk techniques. We have used

surface-anchored crystals of ZIF-8, a MOF with exceptional stability and potential applications in catalysis[27,28]. We used such ZIF-8 crystals with well-defined crystal plane terminations for nanoscale guest–host investigations with formaldehyde, a volatile organic compound (VOC) indoor pollutant, and model reactant for studying $C_1$-molecule conversion mechanisms[29–32]. Using this approach, we found that gas sorption preferentially occurred on high-index crystal planes, such as corners and edges, of only 10 s of nanometers in size within micrometer-sized crystals. Interestingly, we also found high-energy nanodomains within low-energy facets resulting in heterogeneous intra-facet sorption behavior. Furthermore, we observed that the incorporation of defects (i.e., pyrrole ligands) mainly took place at these nano-size high-energy planes and domains on the ZIF-8 surface. These defects led to the structure-sensitive conversion of formaldehyde resembling the "type II" structure sensitivity in supported catalysts, favored on unsaturated sites[1]. Using our in situ nano-spectroscopy technique, two distinct formaldehyde conversion mechanisms on single crystal planes were evidenced. Overall, we found the sorption and conversion performance of the ZIF-8 crystals to be heavily dependent on structure sensitivity phenomena, and we expect the gathered insights and approach to be highly relevant for a wide range of micro-and nanostructured functional materials.

## Results and discussion
### Structure-sensitive formaldehyde sorption
Surface-anchored ZIF-8 microcrystals with well-defined facets were synthesized through a layer-by-layer (LbL) synthesis[33,34]. Pristine ZIF-8

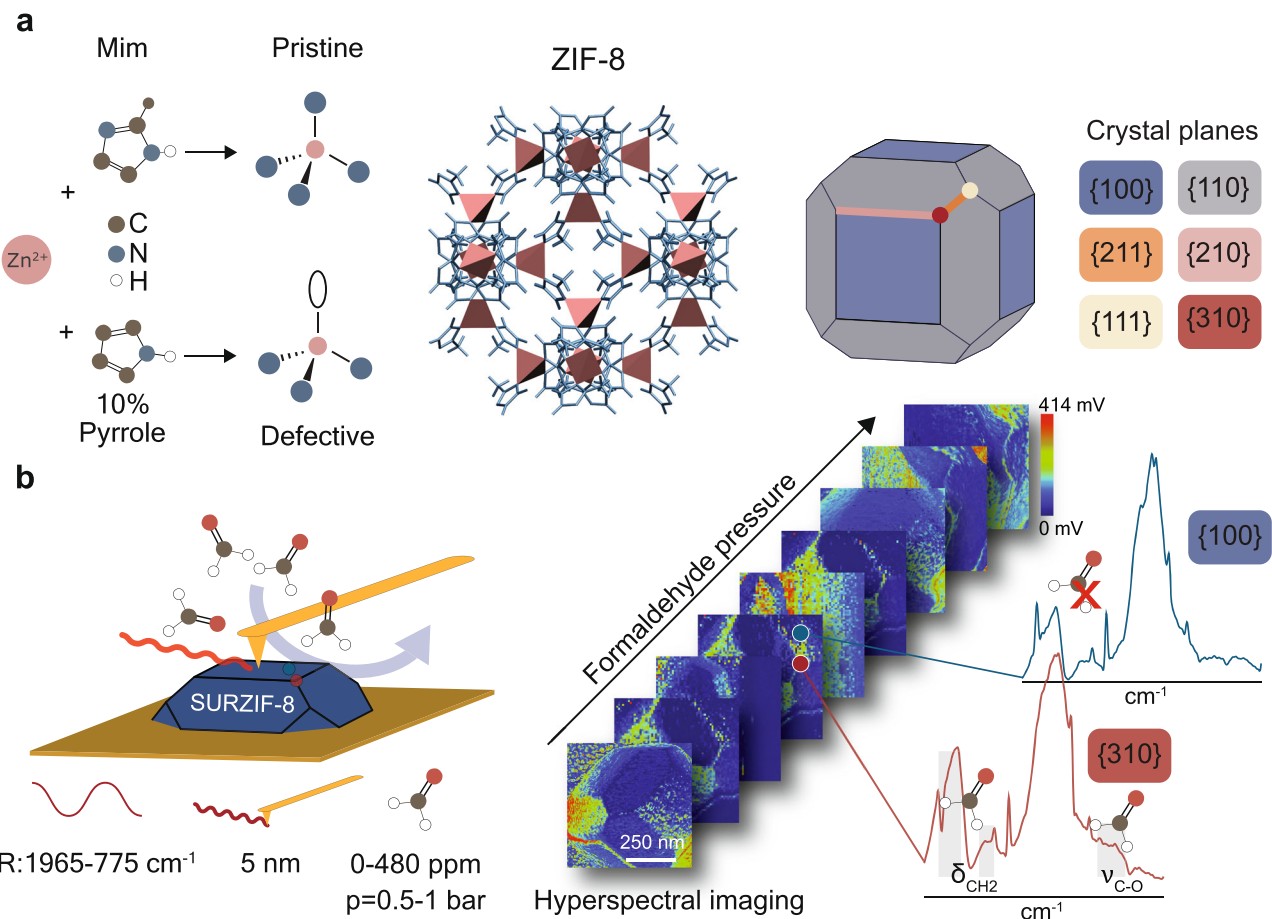

**Fig. 1 | Experimental approach.** Nanoscale guest–host interactions were studied by: **a** Synthesizing (defect-engineered) surface-anchored ZIF-8 microcrystals (Mim: 2-methylimidazole). The structure of ZIF-8 microcrystals is shown on the right, with color-coded crystal plane terminations; **b** Applying in situ photo-induced force microscopy (PiFM), a nano-infrared technique with a spatial resolution down to 5 nm, in the presence of formaldehyde vapor. Using hyperspectral imaging we mapped preferential sorption and conversion sites on ZIF-8 crystals.

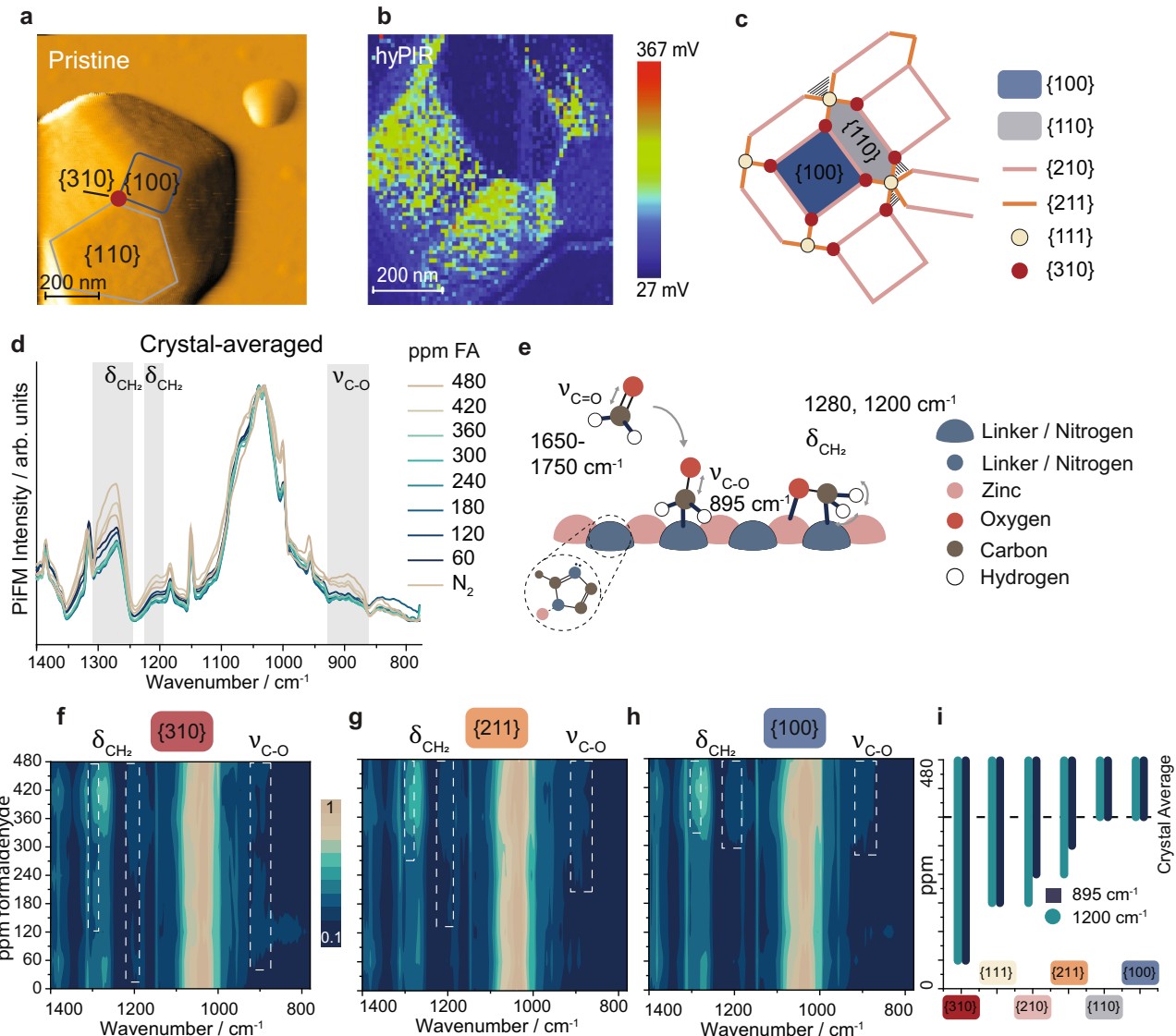

**Fig. 2 | Experimental evidence of structure-sensitive formaldehyde sorption on ZIF-8. a** An atomic force microscopy (AFM) image of ZIF-8 shows the expression of different crystal planes. **b** Hyperspectral images, with a full IR spectrum at every pixel, of such ZIF-8 crystals were recorded during formaldehyde exposure. Overall spectrum intensity per pixel is denoted in mV. Individual planes within these images were analyzed by (**c**) creating morphology-based masks. **d** Crystal-averaged PiFM spectra at increasing formaldehyde (FA) pressure show the increase of three infrared bands, indicating gas sorption. **e** Gas sorption occurs by breaking the C=O bond and forming a covalent $C_{FA}$-$N_{ZIF}$ bond. At the same time, FA is coordinated to the Zn atom through an O-adduct. Contour plots show the formaldehyde sorption-induced change in the IR spectrum for a corner (**f**), an edge (**g**), and a facet plane (**h**). **i** From the contour plots, response pressures to formaldehyde, i.e., the pressure at which a response was recorded, for these crystal planes were found, highlighting the structure-sensitive sorption of formaldehyde.

crystals were synthesized using zinc nitrate and 2-methylimidazole linkers and AFM images showed that the crystals were 0.5–1 micrometer in size and expressed six well-defined crystal plane terminations, namely {100} and {110} facets, {210} and {211} edges, and {111} and {310} corners (Fig. 2a and S10)[35]. Several representative ZIF-8 crystals were used for nanoscale guest–host interaction studies using in situ photo-induced force microscopy (PiFM, Fig. 1b). Hyperspectral (4D) images of the crystals were recorded, composed of a full infrared spectrum (1D) on every pixel (3D: x, y plane, and z height), during stepped gas sorption from 0–480 ppm of formaldehyde (16 vol.% in $H_2O$, Fig. 2b). Since the maximum FA pressure was kept well below the range of critical pressures for gate-opening of ZIF-8 pores, we assume here that gate-opening did not take place during the adsorption of formaldehyde (FA) (Supplementary Fig. 35)[36,37].

The surface-averaged in situ IR spectra for entire ZIF-8 crystals show the sorption of FA on the pristine ZIF-8 surface (Fig. 2d, full spectra and difference spectra found in Supplementary Figs. 18, 19). FA adsorption was observed through the increase of IR bands at 1280, 1200, and 895 $cm^{-1}$, corresponding to $\delta_{CH2, rock}$, $\delta_{CH2, wag}$, and $\nu_{C-O}$, respectively (Fig. 2e)[38,39]. The $\nu_{C=O}$ vibration of physisorbed formaldehyde was not observed as the C=O double bond was broken due to the strong chemisorption of formaldehyde on the ZIF-8 surface (vide infra).

To study the sorption of FA as a function of crystal planes, we masked facet/edge/corner areas of the hyperspectral image based on morphology data (Fig. 2c and Supplementary Figs. 11–15). Subsequently, for each crystal plane, we created contour plots displaying their in situ nano-IR spectra (Fig. 2f–h, Supplementary Figs. 16–19, and Supplementary Table 4). Each of the planes showed the appearance of formaldehyde vibrations. However, the pressure at which these $\delta_{CH2, rock}$, $\delta_{CH2, wag}$ bands appeared, i.e. the response pressure, differed for the different crystal planes (Fig. 2i). This revealed that sorption

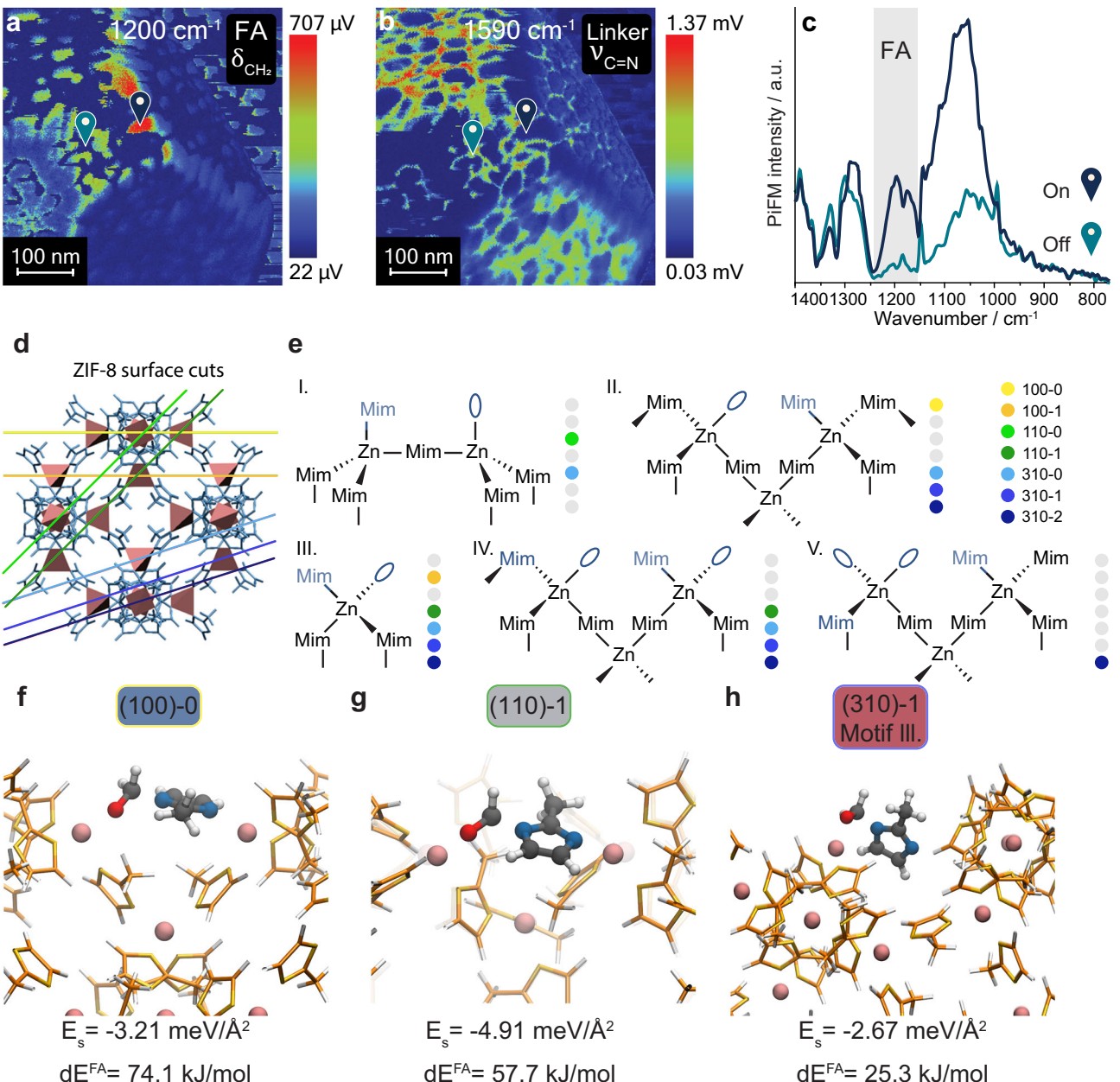

**Fig. 3 | Experimental and theoretical descriptions of heterogeneous gas sorption on various ZIF-8 surface terminations. a** Representative infrared image mapping a chemisorbed formaldehyde vibration (1200 cm⁻¹), showing the presence of nanoislands of alternate sorption behavior within the pristine {100} and {110} ZIF-8 facets. **b** Infrared map of a ZIF-8 framework vibration (1590 cm⁻¹), showing an inverted image, ruling out topography-induced PiFM signal. **c** Point spectra were taken on and off the nanoislands (markers in (**a**, **b**)) showcase non-homogenous sorption behavior, as well as the validity of the IR mapping technique. **d** Crystal surface termination chemistry depends on the orientation and height of the cut of the ZIF-8 crystal, indicated by colored lines (color coding on the right). **e** These terminations can be grouped into surface motifs describing Zn/linker density and positioning, where the surface energy is plane cut-dependent. Density functional theory (DFT) models of (formaldehyde sorption on) the most stable (**f**) {100}, (**g**) {110}, and (**h**) {310} planes show the plane-dependent energy and the corresponding formaldehyde adsorption energy.

occurs first on corners, then edges, and finally on facets in the following order: {310} > {111} > {210} > {211} > {110} ~ {100}. Since all planes can be mapped in a single experiment, we can compare the IR signals of FA on each of these terminations to gain information on the relative strength of adsorption of FA among planes. This demonstrated the structure-sensitive sorption of formaldehyde on ZIF-8 crystal planes, with the expected preference for high-index facets, due to a higher density of unsaturated sites on these planes.

However, in most porous functional materials, isolated and clustered defects are expected to be present in the crystal structure[40]. Therefore, we exploited the nanometer resolution of in situ PiFM to investigate intra-plane heterogeneities in gas sorption behavior. Figure 3a shows a 500 × 500 nm² intensity map of the 1200 cm⁻¹ band, corresponding to chemisorbed formaldehyde, over a pristine ZIF-8 crystal under 300 ppm formaldehyde, which was found to be sufficient for formaldehyde sorption on corners and edges, but insufficient for sorption on facets (Fig. 2i). This IR map evidences sorption on corners and edges, as well as enhanced FA signals on nanosized domains *within* the {100} and {110} planes.

The nanoislands observed in pristine ZIF-8 microcrystals may be explained in three ways: (i) as an artifact induced by overall lower PiFM intensity; (ii) as a result of heterogeneity of ZIF-8 (e.g., a presence of different crystal surface terminations, vide infra); or (iii) as coverage dependent patterns of FA over homogeneous surfaces as a result of interactions between adsorbed FA molecules, similarly to what is observed for e.g., CO adsorption on metal surfaces[41]. To rule out factor (i), we took point spectra in and out of the nanoislands, which showed comparable ZIF-8 vibration intensities, but differences in the FA bands (Fig. 4c and S20). Furthermore, we performed IR mapping of the same area, but with the laser tuned to an aromatic vibration of the ZIF-8 framework ($\nu_{C=N}$ 1590 cm$^{-1}$, Fig. 4b and Supplementary Fig. 21), which showed an inverted image. This showed both the surface sensitivity and validity of the PiFM method and that the external surface is intrinsically heterogeneous, thus supporting hypothesis (ii) over the others (which assume homogeneous surfaces).

To study these heterogeneous planes and to link gas sorption behavior to specific external surface sites within these planes, we modeled three representative ZIF-8 crystallite surfaces using DFT calculations, namely the {100} and {110} facet planes, and a high-index {310} corner plane (Fig. 3d–h). The {100} and {110} planes were chosen because of their high relative crystal surface coverage, and because of the nanoislands observed within these facets. The {310} plane was selected as representative of minority, high-energy terminations showing alternate behavior to formaldehyde vapor. It is crucial to realize that for each of the {100}, {110}, and {310} planes, different surfaces can be constructed by varying the slicing height (Fig. 3d). These surfaces can be grouped into motifs according to the coordination environment of the Zn atom as well as the orientation of terminal ligands (Fig. 3e and Supplementary Fig. 4). These motifs differ in the density of Zn atoms per surface area as well as in the number of Zn-N bonds cleaved, giving rise to various undercoordinated sites. The computed surface energies for the most stable {100}, {110}, and {310} terminations were found to be −3.21, −4.91, and −2.67 meV/Å$^2$, respectively, confirming the higher-energy nature of higher index planes[42].

However, high-energy cuts of, for example, the {100} and {110} planes with energies of 3.45 and 0.62 meV/Å$^2$, respectively, were also found. Because of the moderate differences in surface energies between low and high energy cuts of the {100} and {110} planes, multiple terminations may co-exist on the facet surfaces, for example, as a result of entropic factors. Therefore, the formation of high-energy planes can be favored during synthesis conditions (rather than the 0 K modeling conditions), leading to the expression of the observed nanoislands[43]. Note that negative values for the most stable surface energies were found as a result of the use of a saturated surface model (Supplementary Methods). Using unsaturated surface models did not change our conclusions regarding the relative stabilities of different surface orientations (Supplementary Table 2).

We subsequently modeled FA adsorption on these representative ZIF-8 terminations. In general, we found that the formaldehyde molecules chemisorb on ZIF-8 by forming a carbon adduct with a nitrogen atom from the linker, while the oxygen atom adsorbs on a Zn center, i.e., via the formation of covalent $O_{FA}$-$Zn_{ZIF}$ and $C_{FA}$-$N_{ZIF}$ bonds[5,44]. Such chemisorption is consistent with the absence of molecularly adsorbed formaldehyde in the infrared spectra (Fig. 2d). Additionally, we found that (at 0 K) FA adsorption is an endothermic process (SI). Overall, the trend in FA binding on the most stable cleavages is consistent with experimental findings showing the order {310} > {110} > {100} (Fig. 3f–h and Supplementary Fig. 5). As the most reactive surface termination, we identify motif III, which expresses an isolated Zn site with two terminal linkers, one protonated and one unprotonated (Fig. 3e and Supplementary Fig. 5), which can be found on {310} edges as well as on high-energy cleavages of {100}. Further inspection of the high-energy cleavages showed their stronger FA

binding behavior compared to their low-energy counterparts (e.g., 26.8 versus 74.1 kJ/mol for {100} and 47.1 versus 57.7 kJ/mol for {110}, (Supplementary Fig. 5). Overall, our calculations supported the experimental observations of inter- and intra-plane structure-sensitive sorption, dictated by the formation of nanodomains exposing high surface energy terminations, as described by hypothesis (ii).

**Structure-sensitive defect engineering.** To further explore site-specific FA adsorption, we purposefully incorporated 10% of deuterated pyrrole defect linker in the ZIF-8 crystals to enrich the surface chemistry with undersaturated metal sites (Fig. 1a). Deuterated pyrrole linkers were used because of their distinct spectrum compared to the 2-methylimidazole linkers, which allows one to observe the incorporation of defects with PiFM. Atomic Force Microscopy (AFM) images of the defective crystals showed a change in the expression of its crystal planes: an increase in {100}:{110} facet size ratio was observed (Fig. 4a and Supplementary Figs. 9, 10). This suggested a change in relative plane energies upon defect engineering, which we attribute to strain relaxation of the external surface terminations[45]. These findings are supported by DFT results, which show that an exchange of imidazole linker with pyrrole is thermodynamically most favorable on the high energy cut of the {100} plane (Supplementary Table 3), thereby effectively lowering a difference in surface energies between {100} and {110} planes, resulting in favored {100} expression.

To inspect whether defect linker incorporation can be linked to surface energies, we measured a $1 \times 1\,\mu m^2$ hyperspectral image of a defective crystal in $N_2$. Crystal-averaged IR spectra showed the vibrational fingerprint of pyrrole-$d_5$ by two IR bands at 960 and 885 cm$^{-1}$, corresponding to $\delta_{CD}$ and $\nu_{C=N}$, respectively (Fig. 4b and Supplementary Fig. 22)[46]. Further inspection of the spectral data also revealed the presence of infrared bands at 840 and 790 cm$^{-1}$, corresponding to two $\delta_{OH}$ vibrations of Zn-$OH_2$ sites[47]. These bands indicate the enrichment of the external ZIF-8 surface chemistry with Lewis acid sites (Zn$^{2+}$) and weakly basic sites (Zn-OH) upon the incorporation of pyrrole. We then averaged the spectra of facet, edge, and corner pixels and calculated their 885/1590 cm$^{-1}$ (pyrrole/imidazole) peak ratios. We plotted these ratios together with the crystal-averaged ratio in Fig. 4c. These ratios showed that the expression of defect sites scaled with surface energy and is thus structure sensitive, with the {310}, {210}, and {211} planes having an above-average concentration of defects.

Isotope-labeled deuterated pyrrole was used to pinpoint the distribution and clustering of pyrrole defects using their spectral fingerprint, and to correlate defect-specific guest–host interactions. To visualize pyrrole incorporation, we acquired another $200 \times 200\,nm^2$ hyperspectral images of a defective ZIF-8 crystal in $N_2$. Subsequently, we used principal component analysis (PCA) and clustering of the hyperspectral image, to group pixels into clusters based on spectral similarities (Fig. 4d, Supplementary Information). The spectra of the clusters revealed that clustering mainly occurred based on defect concentration, as all spectra showed a largely unchanged ZIF-8 spectrum yet showed variation in relative $\delta_{CD}$ and $\nu_{C=N}$ band intensities (Fig. 4e and Supplementary Fig. 22).

To deduce the location and degree of clustering of defects, we first sectioned the clustered image into one {100} and two {110} facets, based on the morphology map (Fig. 4d and Supplementary Fig. 22). Then, we calculated the fraction of facet surface that was covered by each cluster (Supplementary Figs. 23–26) which is represented by the bubble sizes in Fig. 4f. Subsequently, we quantified the relative defect linker concentration of each cluster by calculating the peak ratio between pristine and defective ZIF-8 vibrations (i.e., 885/1590 cm$^{-1}$) for the five clusters (Fig. 4f, y-axis). The weighted average of defect linker concentration is also reported per facet (stars in Fig. 4f). In this way, we were able to distinguish between plane-averaged defect concentrations and nanodomains of high/low defect concentration. These results showed that defect concentrations varied more intra-facet, i.e.

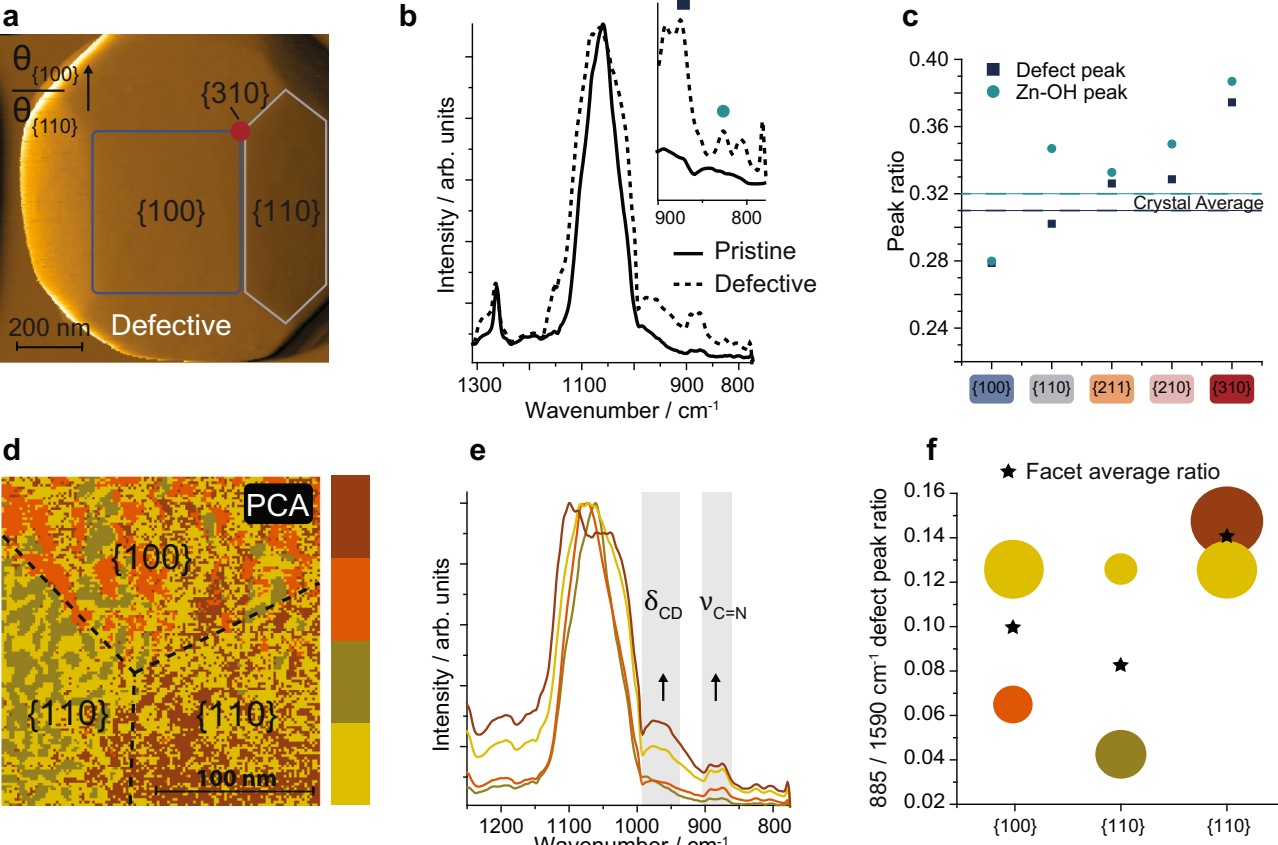

**Fig. 4 | Nanoscale visualization of heterogeneity within crystal planes of ZIF-8.** **a** Representative AFM image of pyrrole-$d_5$ defect-engineered SURZIF-8, showing a higher ratio of {100}/{110} facets compared to the pristine crystals (Fig. 2). **b** Spectroscopic evidence for defect linker incorporation was found when comparing the crystal-averaged spectra of a pristine (solid trace) and a defective (dashed trace) crystal. The defective ZIF-8 spectrum showed $\delta_{CD}$ and $\nu_{C=N}$ vibrations at 960 and 885 cm$^{-1}$ corresponding to the defect linker, and $\delta_{OH}$ vibrations at 840 and 790 cm$^{-1}$ corresponding to defect-induced Zn-OH sites, see inset. **c** Using peak ratios of the defective linker (885/1590 cm$^{-1}$) and Zn-OH (790/1590 cm$^{-1}$) sites, we show that the concentrations of such sites are larger for high-energy planes. Crystal-averaged peak ratios are given by solid lines. **d** A clustered 200 × 200 nm$^2$ hyperspectral image made through principle component analysis (PCA) and clustering where each color refers to a cluster of similar spectral identity, and **e** the corresponding spectra of each cluster of defective ZIF-8 under N$_2$ showing one {100} facet and two {110} facets. Clusters were calculated using principal component analysis (PCA) and clustering, to visualize ZIF-8 defect linker distribution ($\delta_{CD}$ and $\nu_{C=N}$ in **e**). **f** Bubble plot where bubble size corresponds to the fraction of a facet attributed to a certain cluster of the corresponding color, and the bubble position on the y-axis indicates the relative defect concentration. For clarity, the two main contributing clusters were selected for each facet. Facet-averaged defect concentrations are marked with a star. This plot shows differences in relative defect concentrations among facets with comparable surface energies. However, it also shows that strong concentration differences between each facet and their highly localized intra-facet domains can be found.

between parent facet and nanoislands, than inter-facet, an experimental insight that could only be gained because of the nanoscale resolution of the PiFM method. Notably, the defect-rich domains were similar in size to the high-energy termination domains formed on non-defective ZIF-8, which preferentially adsorbed FA also on non-defective ZIF-8 crystals (Fig. 3), suggesting that pyrrole is preferentially incorporated into high-energy planes. DFT calculations on defect incorporation further support this claim (Supplementary Table 3).

**Structure-sensitive formaldehyde conversion.** To discern between the effects of crystal plane terminations and pyrrole defects on the structure-sensitive adsorption of formaldehyde, we performed in situ PiFM FA adsorption experiments on the defective crystals (Fig. 5a and Supplementary Fig. 27). The rising intensity of 1280, 1200, and 895 cm$^{-1}$ bands with FA pressure (0–480 ppm) indicated the sorption of formaldehyde, similarly to the pristine crystals. However, in addition to these bands, signals at 1580, 1380, 1320, 1150, and 1060 cm$^{-1}$ were found. Such bands suggest the formation of different external surface adsorbates, such as formates, dioxymethylene (DOM) and polyoxymethylene (POM), and methoxy

species (Fig. 5c, d and Supplementary Table 4)[32,39]. The systematic occurrence of these adsorbates showed that defect-mediated conversion of formaldehyde was taking place on the defective ZIF-8 surface[32,48]. The results were corroborated by bulk in situ FA adsorption experiments on commercial and synthesized ZIF-8 powders and films, using attenuated total reflection FTIR spectroscopy, in which comparable spectral features and pressure-dependent trends in signal intensity were observed (Supplementary Figs. 36–40).

Based on our spectroscopic results, and in accordance with the literature, we propose two possible formaldehyde conversion mechanisms that operate in parallel on the defective ZIF-8 surface: a methoxy and a formate mechanism (Fig. 5c, d)[49,50]. In the methoxy mechanism, the chemisorbed FA undergoes a Cannizzaro-type reaction to form a methoxy species, a precursor for methanol and formic acid. In the formate mechanism, the FA is chemisorbed in the form of polymers such as dioxymethylene and polyoxymethylene, which are oxidized to form monodentate formates and converted to yield formic acid. Highly relevant for these mechanisms is the hydrolysis of the open zinc sites (vide supra), which provide an additional source of oxygen, which are in turn continuously replenished by water from the

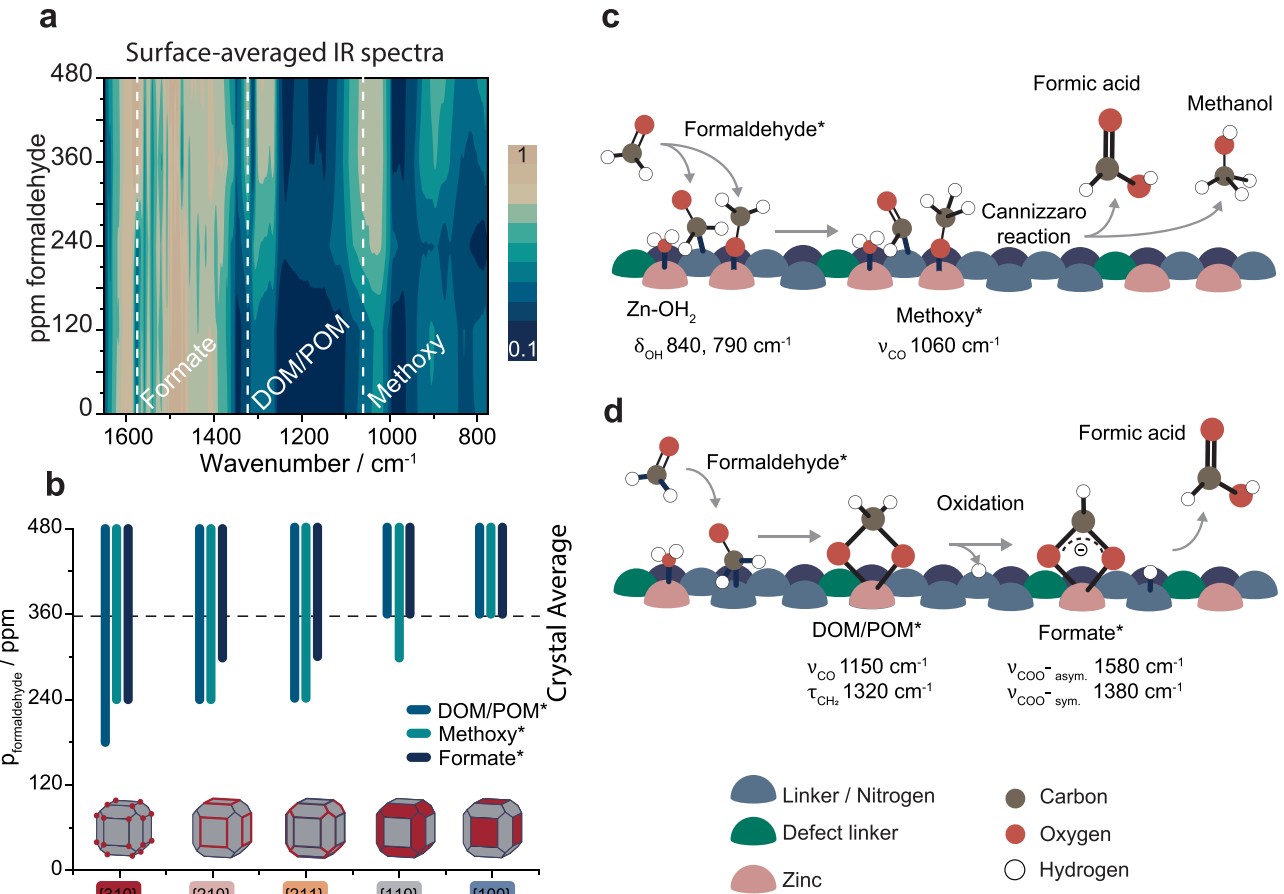

**Fig. 5 | Structure-sensitive conversion of formaldehyde (FA) over defect-engineered ZIF-8. a** Contour plot of the crystal surface-averaged in situ infrared spectra of a defective ZIF-8 crystal, showing the formation of formate, di-/poly-oxymethylene (DOM/POM), and methoxy surface species (relevant bands indicated by dashed lines in the contour plot). **b** A plot showing plane-dependent response pressures for intermediate surface species, thus showing structure-sensitive FA conversion, as intermediates are observed at lower pressures on high-index planes (note: pyrrole-induced changes in crystal facet size ratios prevented the analysis of the {111} facets). Based on the observed species and in accordance with the literature[32,48], we propose two possible mechanisms for FA conversion over defective ZIF-8: a methoxy-mediated (**c**) and formate-mediated (**d**) mechanism.

aqueous formaldehyde solution. These findings are in line with bulk IR studies on oxide-supported metal nanoparticles, where oxygen from the support is instrumental to the conversion of formaldehyde[51,52].

To study the effect of plane terminations (structure sensitivity) on FA conversion, we first compared the response pressure of FA conversion over different crystal planes (Fig. 5b). This comparison showed that FA conversion, similar to FA adsorption, is structure sensitive as a function of crystal termination, with the following order: {310} > {210} > {211} > {110} ~ {100}, which is consistent with what was observed for pristine crystals (Fig. 2). However, by using PCA and clustering within a single facet (e.g., {100}), we showed that defect-rich areas did show more responsive FA (conversion) behavior than the facet average (Supplementary Figs. 28, 29), similarly to what observed for nanoislands on pristine crystals (Fig. 3).

Since different crystal planes showed a different concentration of defects, dependent on the plane energy (Fig. 4c), we subsequently performed a control analysis to rule out a defect concentration effect on the trends observed in Fig. 5b. To this end, we used PCA and clustering to identify defect-rich and defect-poor areas spanning over the entire crystal surface (i.e., a combination of contributions from all planes). Both areas showed comparable FA adsorption and conversion behavior (Supplementary Figs. 28, 30). Therefore, we conclude that the defect concentration did not affect the trends observed in Fig. 5b and that a higher defect concentration did not result in the detection of FA conversion intermediates at lower pressures.

To rationalize these findings, we modeled FA adsorption on the defective, low energy cuts of {100}, {110}, and {310} surfaces by DFT and compared the results to the previously discussed pristine ZIF-8 (Supplementary Fig. 31). Formaldehyde reaction energies (at 0 K) were found to be 97.1, 88.9, and 1.6 kJ/mol for the low-energy cuts of {100}, {110}, and {310}, respectively, again confirming experimental FA sorption (and conversion) trends. The models further showed that the absence of the second nitrogen atom on the pyrrole linker resulted in the formation of undersaturated Zn sites exhibiting Lewis acidity, in contrast to what was observed for pristine sites. We thus confirm that such Lewis acidity was responsible for the subsequent conversion of chemisorbed formaldehyde.

Overall, the present in-depth nano-spectroscopic characterization campaign revealed that: (i) as expected, high-index crystal terminations outperform extended facets in FA adsorption due to exposure of undercoordinated moieties; (ii) seemingly extended and highly coordinated surfaces in MOFs microcrystals contain uncoordinated nano-domains which have similar behavior to high-index terminations, resulting in locally enhanced gas sorption; (iii) the inclusion of defects results in structure-sensitive FA conversion, with higher defect density in high-energy surfaces; (iv) the facet-dependent structure sensitivity of FA adsorption is still observed in defect-engineered microcrystals, meaning that synthesis efforts for better gas sorption materials should be directed to the exposure of high-index crystal planes, rather than to higher defect density. These findings conclusively showed that not all

defect sites are equal, adding a layer of complexity to the structure sensitivity of FA sorption and conversion on ZIF-8, which could only be discovered using the proposed spatially resolved, in situ PiFM technique.

Using a combination of well-defined ZIF-8 microcrystals, nanoscale in situ PiFM measurements and DFT calculations, we have shown the existence and importance of structure sensitivity in formaldehyde sorption and conversion over the porous functional material ZIF-8. We have shown that for (defect-engineered) ZIF-8 formaldehyde gas sorption preferentially occurs on the sterically isolated external surface sites of high-energy planes, such as edges and corners ({310} > {111} > {210} > {211} > {110} ~ {100}), and that defect linker incorporation adheres to this surface energy-based trend as well. Furthermore, as a result of the nanoscale infrared resolution, we found the co-existence of high-energy and low-energy crystal surface terminations within single facets, where the high-energy nanodomains showed enhanced formaldehyde sorption behavior. Similarly, such nanodomains show higher affinity to defect linker incorporation than their low-energy counterparts. Additionally, the incorporation of defects was found to lead to the formation of Lewis acidity and the consequent conversion of formaldehyde on the ZIF-8 surface through both a formate and a methoxy mechanism. Importantly, the performance of these defect sites for formaldehyde conversion was found to be heavily dependent on structure sensitivity phenomena, rather than on defect concentration. We believe that in situ PiFM is a highly applicable analytical toolbox for establishing nanoscale structure-performance relationships since localized sorption behavior on the nanoscale will greatly affect the macroscale performance of (porous) functional materials (e.g., MOFs, zeolites) and guest/probe molecules such as $CO_2$, NO, CO, etc. providing insights for the rational synthesis of improved nanostructured sorbents and catalysts. Outside the field of catalysis, we see high relevance to the understanding of nanostructured functional materials for sensing, gas separation (e.g., composite membranes), and drug delivery, and for environmental science problems such as understanding micro/nano-plastic degradation and atmospheric nanoparticulate chemistry.

## Methods

### Chemicals
The following chemicals were used: 4-mercaptopyridine (MPyr, 95%, Sigma-Aldrich), ethanol (99.5%, Acros), zinc(II) nitrate hexahydrate (98%, Sigma-Aldrich), 2-methylimidazole (99%, Sigma-Aldrich), pyrrole (98%, Sigma-Aldrich), pyrrole-$d_5$ (98 atom% D, Sigma-Aldrich), methanol (99.9%, Sigma-Aldrich), formaldehyde (16% w/v in $H_2O$, Thermo Scientific). Substrates of 60 nm Au on Si, with a 5 nm Ge adhesion layer, were provided by AMOLF.

### Substrate preparation
Gold wafers were cleaned in three successive steps. (1) A photo-resist layer was removed by rinsing the substrate first in acetone, then in deionized water, and lastly in ethanol. (2) The substrates were dried flowing $N_2$ (4 bar), and (3) cleaned using UV-ozone for 20 min. These clean substrates were immersed in 2 mL of 1 mM ethanolic 4-mercaptopyridine solution for 48 h to functionalize the gold surface with nucleation points for ZIF-8 growth. After functionalization, substrates were rinsed in flowing ethanol before drying with $N_2$ (4 bar).

### MOF deposition
Surface-anchored ZIF-8 (SURZIF-8) crystals were grown on a gold substrate using a Layer-by-Layer (LbL) synthesis[1]. Automated LbL synthesis was performed using a Holmarc Successive Ionic Layer Adsorption and Reaction (SILAR) setup. Parameters used were: 100 layers of SURZIF-8 were deposited on the substrate by consecutively dipping the substrate for 2 min in 1 mM methanolic Zn solution and 2 min 2 mM methanolic linker solution. After these deposition steps,

the substrate was rinsed in methanol for 3 s. while stirring at 50 rpm. The substrate was not dried in between cycles and was left to dry in air after the deposition of 100 layers. During synthesis, the solvent was replenished every 30 min. For the defective SURZIF-8 samples, a molar ratio of 10:90 pyrrole(-$d_5$):2-methylimidazole was used, while the overall linker solution concentration was kept constant.

### X-ray diffraction (XRD)
X-Ray Diffraction was performed on a Bruker D8 in grazing incidence diffraction geometry. XRD patterns were recorded in a 2θ range of 4–20 degrees, the incident angle was 0.3 degrees. A Cu source was used.

### Photo-induced force microscopy (PiFM)
Photo-induced force microscopy (PiFM) measurements were performed using a VistaScope from Molecular Vista, Inc. NCHAu25-W AFM tips (Force constant 10–130 N/m, resonance frequency >320 kHz) coated with 25–70 nm of gold by Molecular Vista were used in dynamic non-contact mode. A driving amplitude of 2 nm was used, and a frequency sweep of the cantilever was performed prior to measurements to determine cantilever resonance frequencies. After this sweep, the detection of topography was set to the second mechanical eigenmode resonance ($f_1$), and photo-induced signal detection was set to the first mechanical eigenmode resonance ($f_0$). Determination of the first mechanical mode was repeated after approaching the tip to the surface. By using independent eigenmodes, topography and PiFM signal could be detected simultaneously. Using a cantilever setpoint between 80–85%, 256 × 256-pixel topography images of varying sizes were collected. The midIR source was a Block Engineering tunable quantum cascade laser (QCL) providing a working range of 1965-785 $cm^{-1}$. Averaged IR spectra were taken in sideband mode (pulse modulation: $f_m = f_1-f_0$) with a pulse duration of 32 ns, and with a spectral resolution of 1 $cm^{-1}$. Power levels (iris set between 55–90°, <1 mV of most intense peak during engaged first eigenmode frequency sweep) and acquisition times (between 0.1–1 s) were varied between measurements in such a way that with minimal power, a sufficient signal was collected (>300 μV of the most intense peak in engaged first eigenmode frequency sweep). The approximate lateral resolution of IR spectra was <10 nm. PiFM data were analyzed using SurfaceWorks software.

### In situ PiFM measurements
The procedure for in situ PiFM measurements was as follows. Samples were placed on the PiFM sample stage and were covered by the vacuum cover to create the gas cell. The topography of the crystals was investigated in air, and system quality checks (such as cantilever quality, laser alignment, etc.) were run. Subsequently, the gas cell was purged of air by performing three vacuum/$N_2$ purge cycles. In these cycles, the cell was put under vacuum for 10/5/3 min., refilled with $N_2$, and left to purge for 10/5/3 min for cycles 1–3, respectively. The pressure was then decreased to 0.8 bar $N_2$ to increase sensitivity through an increase in Q factor, while simultaneously preventing air contamination of the gas environment. The absence of water in the gas cell was verified by taking an infrared spectrum in direct drive mode. Due to a change in gas environment/cantilever dampening, a new frequency sweep was performed prior to approaching the surface. Additionally, the IR laser alignment was checked before in situ measurements, and then kept (when possible) unchanged throughout the measurement series to prevent alignment artifacts. Using 256 × 256-pixel topography data, a ZIF-8 crystal was selected for in situ analysis. To verify the ZIF-8 chemistry, IR point spectra were taken (0.3 s. per spectrum, 200 averages). These spectra were then used to optimize the spectrum acquisition time (between 0.1–0.4 s. per spectrum, resulting in 8–30 min hyPIR image measurement time), as well as laser power levels (PiF intensity for the most intense

IR band <1 mV) for the hyperspectral measurements. The image resolution was set to 64 × 64 pixels for the hyPIR measurements to ensure fast data collection and tip quality preservation. All hyPIR images were recorded at set pressures, and not during flow to prevent flow-induced noise and drift. Set pressures of 0, 60, 120, 180, 240, 300, 360, 420, and 480 ppm of formaldehyde were used. Formaldehyde pressure was increased by flowing a stream of nitrogen through a bubbler with 16% w/v formaldehyde in water, thereby also introducing water to the gas environment, with the cantilever retracted 300 micrometers from the surface. During in situ measurements, quality checks were performed by ensuring low amplitude signals, and avoiding topography-induced PiF signal, by adapting feedback parameters- and setpoint values. In case of observed drift during hyPIR measurements, the general procedure was as follows: if the strong drift was observed in the first few lines of a new hyPIR image scan, the measurement was stopped, the ZIF-8 crystal was recentered in the field of view of the AFM image, and the measurement was restarted. This procedure was repeated (where possible) until acceptable levels of drift were obtained. In between, or after hyPIR measurements, more IR spectra and IR maps at relevant wavenumbers were taken.

Hyperspectral data analysis was performed using a combination of SurfaceWorks (Molecular Vista), Gwyddion, and TXM-Wizard software. Masking of hyperspectral images was performed manually for (defective) ZIF-8 crystals at increasing formaldehyde pressure to construct plots shown in Figs. 2f–i, 5b. The masking procedure was as follows: for each hyperspectral measurement, the morphology data were inspected for crystal plane localization (Supplementary Fig. 1a, b). A map detailing crystal plane locations was drawn up (Supplementary Fig. 1c). Individual masks for all individual planes were drawn within the morphology data using a tailor-made Matlab script. Examples for a {110} facet mask, a {210} edge map, and a {111} corner map are given in Supplementary Fig. 1d–f. These maps were overlaid on the hyperspectral image (Supplementary Fig. 1b). Resulting mask-averaged spectra were inspected for outlier behavior. After discarding of outlier data, mask-averaged IR spectra were then further averaged per crystal plane to increase the S/N ratio. These formaldehyde pressure-dependent, plane-averaged IR spectra were subsequently used for contour plot construction and peak ratio calculations.

### Principal component analysis (PCA) and k-means clustering

Principal component analysis (PCA) and k-means clustering was used to analyze the hyPIR images. Prior to PCA and clustering, the data were normalized by sum image division. In the TXM-XANES-Wizard[2], the data were mean-centered prior to PCA and k-means clustering. The appropriate number N of principal components (PCs) was selected upon inspection of the scree plot, Eigenspectra, and Eigenimages. K-means clustering was then performed in N-dimensional PC space using twice the number of clusters (2 × N). The result of k-means clustering was subsequently refined by gaussian mixture modeling (GMM) using expectation maximization (EM). This provided a class membership value (between 0 and 1) of each pixel to each cluster. The spectrum of each cluster was then determined as the weighted average of all pixels using this class membership. This method thus efficiently pools pixels based on spectral similarity (and not on spatial correlation) and provides excellent estimates for the variance in spectral features present in the data.

### Data availability

The authors declare that all data supporting the findings of this study are available within the paper and its supplementary information files. All the raw data generated in this study are available upon request.

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

## Acknowledgements

This work was performed with funding from the European Research Council (ERC) Advanced grant 321140 (B.M.W.) and funding through the Gravity program of the Multiscale Catalytic Energy Conversion (MCEC) Consortium (BMW). Additionally, this work was carried out on the Dutch national e-infrastructure with the support of SURF Cooperative. Furthermore, the authors kindly acknowledge Max Kiffen (Utrecht University, UU) for his help in establishing synthesis parameters for the Layer-by-Layer SURZIF-8 deposition.

## Author contributions

Conceptualization: G.D., M.M., and B.M.W. Methodology: G.D., M.M., K.S., B.B., B.M.W., and F.M. Investigation: G.D., K.S., B.B., and B.M.W. Formal analysis: G.D., M.M., K.S., B.B., B.M.W., and F.M. Software: F.M. Funding acquisition: B.M.W. Project administration: G.D. and B.M.W. Supervision: B.M.W. Visualization: G.D., M.M., K.S., and B.M. Writing—original draft: G.D., M.M., K.S., and B.M.W. Writing—review and editing: G.D., M.M., K.S., B.M., B.B., F.M., and B.M.W.

## Competing interests

The authors declare no competing interests.
