## [Peer Review File · Nature Communications]

Structure Sensitivity in Gas Sorption and Conversion on Metal-Organic FrameworksREVIEWER COMMENTS

Reviewer #1 (Remarks to the Author):

This is a fascinating piece of work, with detailed surface chemistry elucidating nanoscale changes in adsorption properties based on defects (or not as the case may be!), and surface chemistry. I've long believed understanding what goes on at the surface of MOFs during catalytic and adsorption is essential for understanding and tuning their properties, and this is a really nice study detailing exactly this. I particularly liked Figure 2, where nano-IR contour plots showed different responses for the attached aldehyde. I did wonder though, ZIF-8 is prone to gate-opening, with the imidazole rings twisting on exposure to gases and other liquids on increasing guest content. It has also been observed to be crystallite size dependent, with smaller crystallites showing behaviour at very low pressures. How do the authors know that their crystals are not undergoing this phase transition to an open form during formaldehyde uptake? Also in the defective crystals in N₂? I think this is a very important point, as it would also explain differences in nano-IR spectra. The planes cut through would also give rise to different surfaces, once twisted. See the paper by Zhang et al. in J. Phys. Chem (<https://pubs.acs.org/doi/full/10.1021/jp5081466>). I can see that the authors have tried to address this in Figure S35 with the methanol adsorption isotherm. Could more detail be given here? Was this on pristine ZIF-8? What was the particle size? Was any difference observed for the defective ZIF-8? I do think that showing that the concentrations used here are much lower and therefore all at the surface is a fair assumption, but I missed this when initially reading the manuscript, and it's an important assumption. Can the authors comment and make this more obvious in the manuscript?

A few minor comments below;

Page 2, 'in literature' should read 'in the literature'

Page 3 'external surfaces of a MOF crystal planes' should read 'external surfaces of MOF crystal planes'

Page 4 'in situ' should be 'in-situ'

Page 8, the authors state that 'with the expected preference for high index facets.' Why expected? I'm sure I'm missing something obvious? Is this because they are in the pores? Could a sentence be added here?

Could the text to Figure 3 be cut down? For example, the text at the end from 'Furthermore' onwards could be included in the text?

All in all, this is a very comprehensive study with copious volumes of SI. I'd be happy to if the authors can address this question, I'd be very happy to see this paper published. What a lovely piece of work.

Reviewer #2 (Remarks to the Author):

This is a valuable paper for two distinct but equally important reasons. Firstly, the results provide direct insight into the subtleties of a technologically relevant heterogeneous catalytic reaction, the experiments very cleverly isolate the critical features of the surface that contribute to the activity; the surface structural sensitivity first proposed through theory and indirect experimental evidence over bulk materials, then through experiments with idealised single crystal surfaces is now being revealed in situ under conditions where the full complexity can be perceived and explored.

Secondly, the paper is a powerful demonstration of the capability of PiFM, when correctly employed, as an in-situ tool for catalysis research; I predict the paper will become a classic in the catalogue of key works in catalysis research and related work such as gas adsorption and surface treatment.

The field of catalysis has long recognised the importance of in situ measurements, and because of the importance of vibrational spectroscopy in understanding chemistry infrared and Raman spectroscopies have been imaginatively employed to probe reactions in progress. However, what PiFM allows, and this paper by Delen et al. cleverly exploits, is the correlation of molecular identification through vibrational spectroscopy with position at a nanometre scale.

The experimental approach to the problem of formaldehyde adsorption and reaction at ZIF surfaces is thorough and complete. Whilst the vibrational peaks observed in the PiFM are weak, the correlation of position and relative intensity with those of the bulk averaged FTIR-ATR data in

the ESI document provides excellent support for the assignments made. The use of pyrrole to introduce defect sites opens a new window on the defect chemistry of the system with very interesting consequences; the ability to locate nanodomains of structurally distinct areas of the surface has been presaged in other PiFM works including our paper on CePO₄ nanorods surfaces but is demonstrated with much more impact here.

My only comments on the manuscript therefore relate to clarification of some aspects:

Drift. In PiFM the sample can show quite significant drifting underneath the tip, since the timescale of the hyperspectral images on which this study is based can be quite long I'm very surprised drift is not a significant problem. Drift is acknowledged in Fig S15 but rates and compensation methods are not described. Scan times for the hyperspectral images are also not given.

Line 22/23 Abstract. "After incorporation of defective linkers" slightly more expanded explanation of this term would help the reader of the abstract.

Line 27 Abstract. "reminiscent of enzymatic binding sites". This idea is not mentioned again in the paper and I'm not convinced of its accuracy or relevance. The expt observation is that low coordination sites are more important than defects, the specific shape requirements of enzyme catalysis are not being demonstrated here.

Line 78/79 "We show that only data averaged over the entire crystal, including unavoidable defective fractions, is accessible via standard FTIR spectroscopy." This statement is not strictly accurate, and the conclusion is implicit in the poor lateral resolution of far-field infrared.

Figure 4. A, shows "a higher ratio of {100}/{110} facets", I guess the comparison here is with Figure 2, the reader would be helped by that suggestion. I'm also puzzled by the phrase "facet aspect ratio" (Line 236). What does that mean, is it a ratio of area, of maximum length, minimum length, a ratio of min/max length ratios? In Fig S10 (top left image) for example, one line that appears to be used for an aspect ratio calculation appears to be set diagonally across a facet. Please define this term.

Figure 4 C. The Y-axis is a peak ratio, I can't figure out which peaks are being ratioed.

Figure 4D. What do the colours refer too? I can't see an explanation. I presume the same colours are used for the spectra in Figure 4 E but there are 4 spectra and I don't know what those individual spectra link to. I presume these colours then link to the circles in Figure 4 F, but again I don't understand what the colours refer to.

Figures S13 and S14 have the masks for different crystal terminations interspersed. It makes it very confusing. Why can we have #1 {111} and #2-7 {311} in Fig S14 for example?

Figure S15 What is the colour scale referring to? These are hyperspectral images, but presumably are displayed at a single wavenumber?

Figure S17 Has a set of spectra for {310} but there isn't a mask for {310} only for {311}

Figure S19. What are the difference spectra differenced from? Step by step or all back to the spectrum in N2? Please clarify.

Figure S22. A shift in the band at 1070 cm⁻¹ is mentioned and also very evident in Figure 4E. What do these changes signify?

Figure S35 "concentration" misspelt on X axis

Reviewer #3 (Remarks to the Author):

In this manuscript, the authors show the combination of in situ Photo-induced Force Microscopy

(PiFM) with Density Functional Theory (DFT) calculations to study the sorption and conversion of formaldehyde on the external surfaces of well-defined faceted ZIF-8 microcrystals with nanoscale resolution. The following concerns should be considered:

- 1) How is it observed that the incorporation of defects (i.e., pyrrole ligands) take place at these nano-sized high-energy planes and domains on the surface of ZIF-8? How to achieve controlled incorporation of pyrrole ligands?
- 2) The authors propose that in situ nanospectral techniques can be used to prove two different formaldehyde conversion mechanisms on the single crystal plane. However, the authors only point out two mechanisms, and there are no other relevant experiments and representations to prove it in the main text.
- 3) IR spectra obtains the surface-averaged of the entire ZIF-8 crystal, how to obtain the IR spectrum of each crystal plane?
- 4) Could the author explain how to calculate the amount of FA adsorption for each crystal?
- 5) What is the effect of the incorporation defect on the adsorption process? What is the adsorption of ZIF-8 crystals without incorporation defects? Secondly, ligand deletion occurs when synthesizing ZIF-8 crystals, how to determine that the defect is caused by the incorporation of pyrrole?
- 6) This paper uses IR technology to explore the adsorption behavior of each crystal plane of a single crystal, but in practice, it is often the presence of multiple crystals, or multiple unit cells, in this case, will there be other changes?

RESPONSE TO REVIEWERS' COMMENTS

We sincerely thank the three referees for their valuable and very constructive input on our manuscript as well as the related suggestions for revisions, which are now included in the revised manuscript.

Below, you will find:

- in black - the comments of the reviewers
- in blue - our replies to the reviewers' comments
- in green – the additional sentences that are used in the text to address the comments of the reviewers.
- Any changes and additions made are further highlighted in yellow in the enclosed word document of our revised manuscript.

Reviewer #1 (Remarks to the Author):

This is a fascinating piece of work, with detailed surface chemistry elucidating nanoscale changes in adsorption properties based on defects (or not as the case may be!), and surface chemistry. I've long believed understanding what goes on at the surface of MOFs during catalytic and adsorption is essential for understanding and tuning their properties, and this is a really nice study detailing exactly this. I particularly liked Figure 2, where nano-IR contour plots showed different responses for the attached aldehyde.

Response: We sincerely thank the reviewer for their kind words on our paper.

Comment 1: I did wonder though, ZIF-8 is prone to gate-opening, with the imidazole rings twisting on exposure to gases and other liquids on increasing guest content. It has also been observed to be crystallite size dependent, with smaller crystallites showing behavior at very low pressures. How do the authors know that their crystals are not undergoing this phase transition to an open form during formaldehyde uptake? Also in the defective crystals in N₂? I think this is a very important point, as it would also explain differences in nano-IR spectra. The planes cut through would also give rise to different surfaces, once twisted. See the paper by Zhang et al. in J. Phys. Chem (<https://pubs.acs.org/doi/full/10.1021/jp5081466>). I can see that the authors have tried to address this in Figure S35 with the methanol adsorption isotherm. Could more detail be given here? Was this on pristine ZIF-8? What was the particle size? Was any difference observed for the defective ZIF-8? I do think that showing that the concentrations used here are much lower and therefore all at the surface is a fair assumption, but I missed this when initially reading the manuscript, and it's an important assumption. Can the authors comment and make this more obvious in the manuscript?

Response: We agree that ZIF-8 crystals are prone to gate-opening, as we showed in Fig. S35 for methanol. We have now added more details about this topic in the SI and we have updated Figure S35 to show the methanol adsorption isotherms on all the investigated ZIF-8 samples.

As the reviewer points out, the concentrations used in the FA adsorption study are much lower than the ones generally needed for gate-opening, and therefore we *assume* that no gate opening takes place. We realize this point was not clear in our first version of the article, and we now clarified in the main text of the revised article that this is an assumption in the main text.

“Since the maximum FA pressure was kept well below the range of critical pressures for gate opening of ZIF-8 pores, we assume here that gate-opening did not take place during the adsorption of formaldehyde (FA) (Figure S35). [36, 37].”

And in the SI:

“We therefore here assume that no gate opening takes place during our study.”

Moreover, thanks to the reviewer's comment, we now justified this assumption further by running methanol adsorption experiments on all the ZIF-8 crystals prepared on the ATR crystals (Figure S35, updated). Notably, all samples showed comparable methanol isotherms, despite their different morphology and size (see SEM in Fig. S32). Since the commercial sample has a comparable size to the microcrystals synthesized by LbL synthesis (1 μm) used in the PiFM study, we believe the observations can be used to justify our assumption of absence of gate-opening in the FA adsorption study.

Regarding the possible dependence of gate-opening behavior of different surface termination, in the study by Zhang et al. it is reported that surface termination does not affect the gate-opening pressure, which is instead mostly dependent on particle size. Notably, the particle sizes at which strong deviations are observed are much smaller (10s of nm) than the ones used in the current study (100s of nm- μm).

We now cited the study of Zhang in the SI, added the explanation above to the SI, and updated Figure S35 (see below). We thank the reviewer for pointing this out as we believe it made our manuscript clearer and stronger.

Figure S35. Methanol isotherm obtained from ATR spectra recorded on the pristine (red dots) and defective (blue triangles) ZIF-8 synthesized by solvothermal method and a commercial ZIF-8 (black squares) for increasing methanol concentrations, showing the typical S-shaped isotherm due to gate opening of ZIF-8.

Comment 2: A few minor comments below;

Page 2, 'in literature' should read 'in the literature'

Response: The typo was corrected, and similarly in the rest of the paper.

Comment 3: Page 3 'external surfaces of a MOF crystal planes' should read 'external surfaces of MOF crystal planes'

Response: The typo was corrected.

Comment 4: Page 4 'in situ' should be 'in-situ'

Response: While the use of *in-situ* is widespread in the literature, we prefer the use of *in situ*, which is the correct form according to the English dictionary. We nonetheless corrected instances in which the word was not italic in the text.

Comment 5: Page 8, the authors state that 'with the expected preference for high index facets.' Why expected? I'm sure I'm missing something obvious? Is this because they are I the pores? Could a sentence be added here?

Response: We thank the reviewer for pointing this out. What we meant is that the trend is expected due to the presence of more unsaturated sites in high index facets, which preferentially adsorb FA. We added a sentence to clarify what we meant in the text.

"This demonstrated the structure sensitive sorption of formaldehyde on ZIF-8 crystal planes, with the expected preference for high index facets, due to a higher density of unsaturated sites on these planes."

Comment 6: Could the text to Figure 3 be cut down? For example, the text at the end from 'Furthermore' onwards could be included in the text?

Response: We cut the text accordingly and included the sentence in the text.

All in all, this is a very comprehensive study with copious volumes of SI. I'd be happy to If the authors can address this question, I'd be very happy to see this paper published. What a lovely piece of work.

Response: Again, thanks for the reviewer's kind words and for their constructive comments.

Reviewer #2 (Remarks to the Author):

This is a valuable paper for two distinct but equally important reasons. Firstly, the results provide direct insight into the subtleties of a technologically relevant heterogeneous catalytic reaction, the experiments very cleverly isolate the critical features of the surface that contribute to the activity; the surface structural sensitivity first proposed through theory and indirect experimental evidence over bulk materials, then through experiments with idealised single crystal surfaces is now being revealed in situ under conditions where the full complexity can be perceived and explored. Secondly, the paper is a powerful demonstration of the capability of PiFM, when correctly employed, as an in-situ tool for catalysis research; I predict the paper will become a classic in the catalogue of key works in catalysis research and related work such as gas adsorption and surface treatment.

The field of catalysis has long recognised the importance of insitu measurements, and because of the importance of vibrational spectroscopy in understanding chemistry infrared and Raman spectroscopies have been imaginatively employed to probe reactions in progress. However, what PiFM allows, and this paper by Delen et al. cleverly exploit, is the correlation of molecular identification through vibrational spectroscopy with position at a nanometre scale.

The experimental approach to the problem of formaldehyde adsorption and reaction at ZIF surfaces is thorough and complete. Whilst the vibrational peaks observed in the PiFM are weak, the correlation of position and relative intensity with those of the bulk averaged FTIR-ATR data in the ESI document provides excellent support for the assignments made. The use of pyrrole to introduce defect sites opens a new window on the defect chemistry of the system with very interesting consequences; the ability to locate nanodomains of structurally distinct areas of the surface has been presaged in other PiFM works including our paper on CePO₄ nanorods surfaces but is demonstrated with much more impact here.

Response: We sincerely thank the reviewer for their appreciation of our research, and for recognizing the potential impact of the study.

Comment 1: My only comments on the manuscript therefore relate to clarification of some aspects: Drift. In PiFM the sample can show quite significant drifting underneath the tip, since the timescale of the hyperspectral images on which this study is based can be quite long I'm very surprised drift is not a significant problem. Drift is acknowledged in Fig S15 but rates and compensation methods are not described. Scan times for the hyperspectral images are also not given.

Response: Drift is indeed a general problem in scanning probe microscopy (SPM) techniques, and we thank the reviewer for pointing out that more details about this aspect are needed. In our study, no compensation method was used for drift. Drift was indeed observed for some measurements (such as in Fig. S15 as the referee pointed out), however, it did not prevent us from identifying the individual crystal planes for analysis. In our experiments, to minimize drift as much as possible, we only scanned our samples after the chamber was filled with the desired gas atmosphere. We therefore used stepped formaldehyde pressures and acquired spectra under steady state, instead of continuously flowing formaldehyde. In the Materials and Methods section of the SI this was previously written down as "All hyPIR images were recorded at set pressures, not during flow to prevent flow-induced noise." This has now been expanded on.

Despite these efforts, drift would at times be observed. The general procedure in case of drift was as follows: if strong drift was observed in the first few lines of a new hyPIR image scan, the measurement was stopped, the ZIF-8 crystal was recentered in the field of view of the AFM image, and the measurement was restarted. This procedure was repeated (where possible) until acceptable levels of drift were obtained. This approach was feasible in our experiments since set pressures of formaldehyde were used, and since the hyPIR measurement times were acceptably short (8-30 min when using a spectral acquisition time of 0.1-0.4 s, respectively). Typically, a spectral acquisition time of 0.3 sec was used to best compare with parameters used for point spectra (see SI), thus resulting in a hyPIR measurement time of 20 min.

To clarify these experimental details, we added the following in the Materials and Methods section of the SI:

- "All hyPIR images were recorded at set pressures, not during flow to prevent flow-induced noise and drift."
- "These spectra were then used to optimize the spectrum acquisition time (between 0.1-0.4 s. per spectrum, resulting in 8-30 min hyPIR image measurement time), as well as laser power levels (PiF intensity for the most intense IR band <1 mV) for the hyperspectral measurements."
- "In case of observed drift during hyPIR measurements, the general procedure was as follows: if strong drift was observed in the first few lines of a new hyPIR image scan, the measurement was stopped, the ZIF-8 crystal was recentered in the field of view of the AFM image, and the measurement was restarted. This procedure was repeated (where possible) until acceptable levels of drift were obtained."

Comment 2: Line 22/23 Abstract. "After incorporation of defective linkers" slightly more expanded explanation of this term would help the reader of the abstract.

Response: We changed the text and added an explanation of defective linkers accordingly in the abstract:

"Additionally, defective ZIF-8 crystals with undersaturated metal centers were prepared by incorporating 10% of pyrrole linkers. On these defective crystals, structure sensitive conversion of formaldehyde through a methoxy- and a formate mechanism mediated by Lewis acidity was found."

Comment 3: Line 27 Abstract. "reminiscent of enzymatic binding sites". This idea is not mentioned again in the paper and I'm not convinced of its accuracy or relevance. The expt observation is that low coordination sites are more important than defects, the specific shape requirements of enzyme

catalysis are not being demonstrated here.

Response: We accordingly removed this piece of sentence.

Comment 4: Line 78/79 “We show that only data averaged over the entire crystal, including unavoidable defective fractions, is accessible via standard FTIR spectroscopy.” This statement is not strictly accurate, and the conclusion is implicit in the poor lateral resolution of far-field infrared.

Response: We have removed this statement from the text as we agree with the referee.

Comment 5: Figure 4. A, shows “a higher ratio of {100}/{110} facets”, I guess the comparison here is with Figure 2, the reader would be helped by that suggestion. I’m also puzzled by the phrase “facet aspect ratio” (Line 236). What does that mean, is it a ratio of area, of maximum length, minimum length, a ratio of min/max length ratios? In Fig S10 (top left image) for example, one line that appears to be used for an aspect ratio calculation appears to be set diagonally across a facet. Please define this term.

Response: We agree with the referee, and have added the following sentence to the caption:

“a higher ratio of {100}/{110} facets compared to the pristine crystals (Fig.2)”

Regarding the “facet aspect ratio”, we agree that the definition was not clear, and we accordingly introduced it in Figure S10. We now refer to the “size ratio” between the facets in the main text.

“Insets are the ratio between the length of the {100} square edge and the length of the adjacent {110} facet parallel to the used square edge. Upon incorporation of 10% pyrrole linker, the size ratio increases.”

Comment 6: Figure 4 C. The Y-axis is a peak ratio, I can’t figure out which peaks are being ratioed.

Response: We thank the reviewer for pointing this out, and we added the corresponding peak used for the ratio of both peak ratios (1590 cm^{-1}). The text was modified as follows:

“Using peak ratios of defective linker ($885/1590\text{ cm}^{-1}$) and Zn-OH ($790/1590\text{ cm}^{-1}$) sites, we show that the concentrations of such sites are larger for high energy planes.”

Comment 7: Figure 4D. What do the colours refer too? I can’t see an explanation. I presume the same colours are used for the spectra in Figure 4 E but there are 4 spectra and I don’t know what those individual spectra link to. I presume these colours then link to the circles in Figure 4 F, but again I don’t understand what the colours refer to.

Response: This was now clarified in the caption. Indeed, in Figure 4F there are only two colours per facet, which corresponds to the main clusters contributing to the facet. The text is as follows:

“A clustered $200\times 200\text{ nm}^2$ hyperspectral image where each color refers to a cluster of similar spectral identity and (E) the corresponding spectra of each cluster of defective ZIF-8 under N_2 showing one {100} facet and two {110} facets. Clusters were calculated using Principal Component Analysis (PCA) and clustering, to visualize ZIF-8 defect linker distribution (δ_{CD} and $\nu_{\text{C=N}}$ in E). (F) bubble plot where bubble size corresponds to the fraction of a facet attributed to a certain cluster of corresponding color, and the bubble position on the y-axis indicates the relative defect concentration. For clarity, the two main contributing clusters were selected for each facet.”

Comment 8: Figures S13 and S14 have the masks for different crystal terminations interspersed. It makes it very confusing. Why can we have #1 {111} and #2-7 {311} in Fig S14 for example?

Response: We understand the confusion, but the figures were generated automatically by the PCA and clustering software. Since the labels generated were then used for further analysis, we prefer keeping the numbers as they are.

Comment 9: Figure S15 What is the colour scale referring to? These are hyperspectral images, but presumably are displayed at a single wavenumber?

Response: We thank the reviewer for pointing this out" we added an explanation accordingly: "Scalebars reflect the IR signal intensity of the most intense IR band within each spectrum."

Comment 10: Figure S17 Has a set of spectra for {310} but there isn't a mask for {310} only for {311}

Response: The reviewer is correct, there was a typo: {311} should be {310} in the masks reported in Figure S14.

Comment 11: Figure S19. What are the difference spectra differenced from? Step by step or all back to the spectrum in N2? Please clarify.

Response: The spectra were differenced against N2. This is now clarified in the caption.

Comment 12: Figure S22. A shift in the band at 1070 cm^{-1} is mentioned and also very evident in Figure 4E. What do these changes signify?

Response: Since both pyrrole and 2-methylimidazole linkers absorb at this wavelength, we do not feel comfortable commenting on the reason for the observed shift, and we deleted the sentence accordingly.

Comment 13: Figure S35 "concentration" misspelt on X axis

Response: We now corrected the typo and updated the figure and added additional data, in accordance with the comments by reviewer 1.

Reviewer #3 (Remarks to the Author):

In this manuscript, the authors show the combination of in situ Photo-induced Force Microscopy (PiFM) with Density Functional Theory (DFT) calculations to study the sorption and conversion of formaldehyde on the external surfaces of well-defined faceted ZIF-8 microcrystals with nanoscale resolution.

Response: We sincerely thank the reviewer for their appreciation of our research, and for the very valuable comments and suggestions for revisions to strengthen the scientific message of our work.

The following concerns should be considered:

Comment 1: How is it observed that the incorporation of defects (i.e., pyrrole ligands) take place at these nano-sized high-energy planes and domains on the surface of ZIF-8? How to achieve controlled incorporation of pyrrole ligands?

Response: To observe the incorporation of defects with PiFM, we have used deuterated pyrrole ligands, which have a distinct spectrum compared to regular, non-isotopically labeled pyrrole, and compared to the 2-methylimidazole linkers (Fig. 4B). In this way we were able to observe where the pyrrole ligands were located on the facets. The defects were preferentially incorporated in high index planes (Figure 4C), and were preferentially clustered in nanosized domains on {100} and {110} facets (Figure 4D, S22). These domains were similar in size to the ones formed on non-defective ZIF-8 due to different possible terminations of the crystal lattice, which preferentially adsorbed FA also on non-defective ZIF-8 crystals (Figure 3). Moreover, DFT calculations supported the observation (Table S3). In view of these observations, we propose that pyrrole ligands clusters coincide with high-energy, undercoordinated domains in the defective ZIF-8. This is now clarified in the text. "Deuterated pyrrole linkers were used because of their distinct spectrum compared to the 2-methylimidazole linkers, which allows one to observe the incorporation of defects with PiFM."

"Notably, the defect-rich domains were similar in size to the high energy termination domains formed on non-defective ZIF-8, which preferentially adsorbed FA also on non-defective ZIF-8 crystals

(Figure 3), suggesting that pyrrole is preferentially incorporated into high-energy planes. DFT calculations on defect incorporation further support this claim (Table S3).”

While achieving controlled incorporation of pyrrole ligands was not our aim in this study, we believe gaining control on the distribution of defects will be challenging. Possibly, ligand exchange strategies could be followed.

Comment 2: The authors propose that in situ nanospectral techniques can be used to prove two different formaldehyde conversion mechanisms on the single crystal plane. However, the authors only point out two mechanisms, and there are no other relevant experiments and representations to prove it in the main text.

Response: Both methoxy- and formate-mediated mechanisms of formaldehyde were previously proposed in the literature. For example, from extensive studies of formaldehyde sorption on metal oxide surfaces studied with bulk IR spectroscopy. [Busca, G., Lamotte, J., Claude Lavalley, J. & Lorenzelli, V. FT-IR study of the adsorption and transformation of formaldehyde on oxide surfaces. *J. Am. Chem. Soc.* 109, 5197–5202 (2002).] In our work, we observed that all crystal planes of the defective ZIF-8 crystals form the intermediates associated with both the methoxy-mediated and the formate-mediated mechanisms. These intermediates, consistent with the mechanisms described in the literature, were only observed on defective ZIF-8 crystals and not on pristine crystals. We therefore pose that defective crystals facilitate the conversion of formaldehyde whereas pristine crystals do not. Additionally, in Fig. S28, we show that regions with high concentration of defects react with FA at lower pressures.

Proving the existence of a certain mechanism is therefore beyond the scope of our study, and we would like to note that a mechanism cannot per-se be proven (see “The Burden of Disproof” by Susannah L. Scott, *ACS Catal.* 2019, 9, 5, 4706–4708). Since these mechanisms were already proposed in the literature, we here aim to show preferential, defect-site induced formaldehyde conversion. Of course, we do hope that future new studies may further substantiate the different routes and maybe lead to new insights that proof or disproof the different mechanisms put forward.

Comment 3: IR spectra obtains the surface-averaged of the entire ZIF-8 crystal, how to obtain the IR spectrum of each crystal plane?

Response: This is a good point and we believe that the reviewer refers here to the limitation of traditional IR spectroscopy techniques, which are limited by diffraction limits of IR light, and therefore would indeed give an average spectrum of entire ZIF-8 crystals or even multiple ZIF-8 crystals. Using PiFM, we were not limited by the diffraction spot of IR, since the tip acts as an antenna for IR light, amplifying the signal from a nm-size spot on the sample. We were thus able to obtain an IR spectrum for every pixel of the microscopy images reported in the paper, together with the corresponding topographic image of the same crystal. This was crucial, since it allowed us to describe both the surface-averaged spectrum of the ZIF-8 crystal surface, by averaging the IR spectra of all these pixels, as well as allow us to separate spectral contributions of different crystal planes. To do so, we used the topography map, which was acquired simultaneously with the IR image, to create masks corresponding to a crystal plane, and finally averaged spectra in such regions (see e.g., Fig. S1, S13, S14) in the SI of the paper. The text above is now included in the SI for clarity.

Comment 4: Could the author explain how to calculate the amount of FA adsorption for each crystal?

Response: The present study aims to show the trends in preferential adsorption of FA on different facets of ZIF-8 crystals. This is shown by the increase in relative intensity of IR signals associated with FA adsorption at different FA pressures for different crystal planes (see e.g. Fig. 2I). Since all planes can be mapped in a single experiment, we can compare the IR signals of FA on each of these planes to gain information on the *relative* strength of adsorption of FA among planes. However, we do not have direct evidence of the absolute amount (e.g., micromoles) of FA adsorbed on each crystal, since

this would require knowing the extinction coefficient of each FA signal, which would in turn require extensive studies beyond the scope of the paper. To clarify this aspect, we added a comment in the main text on the fact that the observed trends are relative trends.

“Since all planes can be mapped in a single experiment, we can compare the IR signals of FA on each of these terminations to gain information on the relative strength of adsorption of FA among planes.”

Comment 5: What is the effect of the incorporation defect on the adsorption process? What is the adsorption of ZIF-8 crystals without incorporation defects? Secondly, ligand deletion occurs when synthesizing ZIF-8 crystals, how to determine that the defect is caused by the incorporation of pyrrole?

Response: The effect of the incorporation of defects on the adsorption process is described in Figure 4 and in the main text, where we show that only after incorporation of pyrrole ligands FA is converted to methoxy and formate species on the surface of ZIF-8 crystals. The adsorption of FA on non-defective ZIF-8 crystals is described in Figure 2 and 3, where we show that high-index surfaces preferentially adsorb FA at lower pressures, and that nanodomains exist on extended {100} and {110} surfaces which expose undercoordinated sites and are capable of adsorbing FA at lower pressures than the average corresponding facet. Regarding ligand deletion, we believe the reviewer is referring to the creation of unsaturated Zn sites upon the introduction of pyrrole ligands. This is shown in Figure 4C: in defective ZIF-8 crystals, we observe IR signals ascribed to Zn-OH (790 cm^{-1}), and these correlate with the signal of deuterated pyrrole ligands (885 cm^{-1}).

Comment 6: This paper uses IR technology to explore the adsorption behavior of each crystal plane of a single crystal, but in practice, it is often the presence of multiple crystals, or multiple unit cells, in this case, will there be other changes?

Response: This is an important observation by the referee. As the reviewer rightly points out, in this study we have used *in situ* PiFM to show the FA adsorption behavior on different crystal planes in single ZIF-8 crystals. In practice, multiple crystals are present in real-world applications, so that the ensemble of crystals will determine the performance of the sample at the macroscale. Nonetheless, the performance of the ensemble will be the sum of the performance of each single crystal, so that if the observed single crystals are representative of the ensemble, the nanoscale observation is relevant and can be used to describe the macroscopic behavior of ZIF-8 samples. To show that the analyzed crystals are representative of the ensemble, we took numerous images of the crystals of both pristine and defective ZIF-8 (see e.g., Fig. S10), showing that the crystals of each sample are comparable in size and relative facet exposure. We would also like to add that the current PiFM approach could be used to describe the behavior of samples which are instead composed of crystals having different shapes, distinguishing families of crystals, or outliers, and their behavior. The following sentence was added to the paper:

“Several representative ZIF-8 crystals were used for nanoscale guest-host interaction studies using *in situ* Photo-induced Force Microscopy (PiFM, Figure 1B).”

“We believe that *in situ* PiFM is a highly applicable analytical toolbox for establishing nanoscale structure-performance relationships since localized sorption behavior on the nano-scale will greatly affect the macroscale performance of (porous) functional materials (e.g. MOFs, zeolites) and guest/probe molecules such as CO₂, NO, CO, etc. providing insights for the rational synthesis of improved nanostructured sorbents and catalysts.”

REVIEWERS' COMMENTS

Reviewer #2 (Remarks to the Author):

I have reviewed the author's replies to the questions from the referees and I am happy with the answers and modifications made. The paper should now be published as it stands.

Reviewer #3 (Remarks to the Author):

The authors have worked carefully through the reviewer comments and suggestions, and I am satisfied with their revisions and replies. This paper seems ready for publication.

I think the authors have answered well on this key point proposed by reviewer #1.